# AesthetiX-RAG: Causally-Grounded Emotion Recognition and Explanation in Paintings via Artist–Style Knowledge and Faithful Visual Evidence

## Abstract

Art provides a visual medium for emotional expression. In paintings, such expression is conveyed through compositional structure, symbolic elements, and stylistic features. However, existing computational methods for understanding artwork often leverage semantic content and low-level visual features. Consequently, these methods may provide a limited representation of emotional expression embedded in stylistic and compositional features. In this work, we present AesthetiX-RAG, a causally grounded retrieval-augmented framework for emotion recognition and explanation in paintings. The proposed framework employs an Artist–Style–Motif–Emotion (ASME) graph to model relationships among artists, stylistic traditions, symbolic motifs, and emotional expression. The artist–style priors derived from ASME are projected into control tokens and fused with visual representations through Multi-Head Attention. The fused representation is used to predict the emotion label. Finally, the retrieval-augmented generator combines the emotion label with faithful visual evidence and retrieved artist–style knowledge to generate a grounded natural-language explanation for the predicted emotion. We also introduce a new dataset AesthetiX-5K, to support emotion recognition and explanation in paintings. The dataset contains 5116 paintings comprising 27 artistic styles, 23 artists, and 10 genres, with each sample annotated with an emotion label and a human-written rationale. Detailed experimental analysis on AesthetiX-5K and existing art-emotion datasets validates the effectiveness of the proposed framework. The code and dataset will be made publicly available.

## 1 Introduction

Art communicates human emotions through the interaction of color, symbols, objects, and compositional structure. Across artistic traditions such as Expressionism and Japanese ukiyo-e, paintings often express emotional meaning through stylistic and compositional features in addition to visible semantic content. Existing computational methods are effective in recognizing objects and generating semantic descriptions. However, modeling human emotional responses to paintings remains challenging because emotional interpretation is not determined by semantic content only. For example, a method may generate an semantic description such as "a woman near a window," but cannot explain how diffused lighting, a subdued color palette, and a downcast gaze contribute to perceived sad emotion. These observations require a framework that models the combined contribution of semantic content, stylistic features, symbolic elements, and compositional structure in perceived emotion.

We propose AesthetiX-RAG, a multimodal reasoning framework for emotion recognition and explanation in paintings. Existing vision–language models extract features related to semantic content. In comparison, the proposed framework combines semantic, symbolic, stylistic, and contextual information for emotion reasoning. This allows the framework to recognize the perceived emotion of a painting and explain how visual evidence, artist–style knowledge, and compositional structure contribute to the emotional interpretation.

AesthetiX-RAG is a framework that integrates visual representation learning, structured art-historical knowledge, causal reasoning, and retrieval-grounded explanation. It comprises four key components: (i) a **Tri-**

**Branch Visual Encoder**, (ii) an **Artist-Style-Motif-Emotion (ASME) Graph**, (iii) a **Counterfactual Style Intervention** module, and (iv) a **Retrieval-Augmented Generator (RAG)** module. The *Tri-Branch Visual Encoder* extracts complementary stylistic, semantic, and iconographic representations from the input painting. The *ASME* graph models dependencies among artists, stylistic traditions, symbolic motifs, and emotional expression, while reducing shortcut correlations through regularized graph learning. The *Counterfactual Style Intervention* module perturbs style-related attributes such as texture, color, and composition to evaluate the consistency of predicted emotions under controlled visual changes. Finally, the *RAG* module combines faithful visual evidence with retrieved artist-style knowledge to generate grounded natural-language explanations. Thus, AesthetiX-RAG embeds explanation within the emotion prediction pipeline rather than using it as a post-hoc component.

To train and evaluate the proposed framework, we construct AesthetiX-5K, a curated dataset containing 5116 painting images across 27 artistic styles, 23 artists, and 10 genres. Each painting is associated with a dominant emotion label and a human-written rationale explaining the corresponding emotional interpretation. In addition to emotion recognition, AesthetiX-5K supports the evaluation of explanation faithfulness through metrics such as Citation Coverage, Attribution Precision, and Non-Hallucination Rate. By coupling fine-art annotations with affective rationales, AesthetiX-5K provides a dedicated benchmark for multimodal emotion reasoning in paintings.

**The main contributions of this work are summarized as follows:**

- **Unified Multimodal Reasoning Framework.** We propose AesthetiX-RAG, a causally grounded multimodal framework for emotion recognition and explanation in paintings. The proposed framework integrates visual representation learning, structured artist–style knowledge, counterfactual style analysis, and retrieval-grounded language generation within a unified reasoning pipeline.

- **AesthetiX-5K Benchmark Dataset.** We introduce AesthetiX-5K, a curated dataset containing 5116 paintings across 27 artistic styles, 23 artists, and 10 genres. Each painting is annotated with a dominant emotion label and a human-written rationale. The dataset provides a dedicated benchmark for multimodal emotion reasoning and explanation faithfulness in paintings.

- **Causal Perspective for Emotion Understanding in Paintings.** We formulate emotion understanding in paintings as a causal reasoning problem. Semantic content, stylistic features, symbolic elements, and compositional structure are modeled as complementary contributors to perceived emotion. This formulation helps examine whether the predicted emotion remains consistent under controlled changes in style-related attributes.

In addition to technical contributions, this work supports the computational understanding of human creativity and artistic expression. By connecting emotion prediction with visual evidence, artist-style knowledge, and cultural context, the proposed framework provides a more transparent, accessible, and culturally informed understanding of paintings. This direction aligns with the broader context of the United Nations Sustainable Development Goals, particularly SDG 10 on Reduced Inequalities and SDG 16 on Peace, Justice, and Strong Institutions. It supports inclusive access to cultural interpretation and explainable AI-assisted engagement with artistic heritage.

## 2 Background and Related Work

**Perceptual Understanding of Paintings.** Existing computational approaches to painting analysis have primarily focused on artist, style, genre, and object-level recognition. Large-scale datasets such as WikiArt (Saleh & Elgammal (2015)), OmniArt (Strezoski & Worring (2018)), and IconArt (Gonthier et al. (2018)) provide large-scale analysis of paintings with stylistic and contextual annotations. Gonthier et al. (2018) proposed a weakly supervised framework for object detection in paintings. Tan et al. (2016) explored deep representations for fine-art classification, while Gairola et al. (2020) learned unsupervised style embeddings for retrieval and recognition. Hierarchical modeling of style relations and robustness across movements were proposed by Menis-Mastromichalakis et al. (2020) and Springstein et al. (2024), respectively. These methods

show that perceptual and stylistic features can be effectively learned from paintings, but the focus on visual appearance rather than emotional interpretation.

**Affective and Emotional Analysis of Paintings.** Building on perceptual models, the subsequent works explored how emotional and aesthetic visual compositions shape responses. The ArtEmis v2.0 (Mohamed et al. (2022)), EmoSet (Yang et al. (2023)), PARA (Yang et al. (2022)), D-ViSA (Kim et al. (2023)), AMT (You et al. (2016)) and BAID (Yi et al. (2023)) datasets provide large-scale annotations of paintings with balanced emotion labels, perceptual features, demographic features, and stylistic context for affective and aesthetic analysis. Utilizing these datasets, Yanulevskaya et al. (2008) and You et al. (2016) modeled emotional valence from visual features to link image composition with affective response. Alameda-Pineda et al. (2019) and Mohamed et al. (2022) introduced multimodal and contrastive frameworks for emotion recognition. Yang et al. (2022), Yi et al. (2023), and Kim et al. (2023) focused on personalized, style-aware, and dimensional emotion modeling in art. Recent multimodal approaches, such as AesExpert (Huang et al. (2024)), employ vision–language models for aesthetic reasoning. These works show a shift from affect labeling to aesthetic reasoning. However, these methods identify correlations between semantic content and emotion without modeling the perceptual basis of affect.

**Vision–language and Large Multimodal Models for Art Interpretation.** Language grounding has expanded art understanding from recognition to interpretation. Bai et al. (2021) proposed a retrieval-augmented captioning model using external knowledge to describe a painting's content, form, and context. Hayashi et al. (2024) introduced an artwork-explanation task assessing how LVLMs integrate visual and textual features. KALE (Jiang et al. (2024)), ArtGraph (Castellano et al. (2023)), GalleryGPT (Bin et al. (2024)) and ArtSeek (Fanelli et al. (2025)) fine-tuned multimodal models such as BLIP-2 (Li et al. (2023)), LLaVA (Liu et al. (2023)), and GIT-2 (Wang et al. (2022)) for interactive interpretation and retrieval-augmented reasoning. Takahashi et al. (2024) applied label-distribution learning for art emotion analysis, while Ramos et al. (2025) used diffusion models to synthesize pseudo-annotations. The ArtEmis speaker model (Achlioptas et al. (2021)) and its contrastive extension Mohamed et al. (2022) linked visual perception with affective text but produced affective text without revealing why specific visual features evoked emotion and lacked contextual grounding in artist or style. Our AESTHETIX-RAG integrates tri-branch visual reasoning, artist–style–emotion graph conditioning, and retrieval-augmented explanation generation to move from emotion description to emotion justification.

**Retrieval-Augmented and Knowledge-Grounded Art Reasoning.** Recent research has explored the use of external knowledge to improve artwork understanding and interpretation. Visual Narratives Springstein et al. (2024) analyzed relationships between artistic style and visual content, while Dynamic Temporal Gating Networks Lee et al. (2025) and AesExpert (Huang et al. (2024)) focused primarily on aesthetic alignment and preference modeling. In a similar way, the ArtRAG Wang et al. (2025) framework incorporated information about the artist, movement, and motif via an Art Context Graph. However, it primarily relies on metadata and explicitly models perceived emotion or generates emotion-grounded explanations.

## 3   Dataset: The AesthetiX-5K Corpus

Understanding emotions in an art piece is different from a conventional vision task as it is not determined by pixels, texture, objects, and color. It also requires reasoning over symbolism, lighting, stylistic texture, and cultural context. To support research in this direction and the development of AesthetiX-RAG, we curated a dataset containing 5116 paintings collected from open repositories. Also, each artwork is annotated with the artist, style, dominant emotion, and a human-written rationale for explanation. Figure 1 presents a selected sample of the dataset.

### 3.1   Dataset Composition and Scope

The dataset contains three features of artistic expression, namely, style, genre, and emotion. It includes the work of popular and influential artists such as Picasso Monet, Rembrandt, Eisen, Monet etc. AesthetiX-5K covers 27 artistic styles like Expressionism, Impressionism, Cubism, etc. It further includes 10 primary genres such as portrait, landscape, still life, cityscape and religious painting. Each painting is annotated

with one of six emotion categories defined by classical affect theory: anger, fear, disgust, sadness, surprise and happiness. Further, each painting includes a human-written rationale consisting of two to four sentences describing how visual or symbolic features in the composition substantiate the annotations and express the intended emotion. These explanations provide interpretive depth and enable affective understanding that bridges formal aesthetics and perceptual response.

## 3.2 Annotation Protocol

A team of three annotators trained in visual arts and psychology annotated the dataset in two phases. In Phase 1, the annotators viewed the paintings without any metadata and selected the dominant emotion from six[1] categories defined by classical affect theory established by Paul Ekman ( Ekman et al. (1999)). In Phase 2, the annotators were provided with the artist and style associated with each painting and wrote a short explanation beginning with *This painting evokes ... because ....* The conflict in labeling were removed by majority voting and textual explanations were merged via grammar normalization and grammar regularization. The inter-annotator agreement (Fleiss' $\kappa$=0.72) indicates good degree of consistency among annotators.

## 3.3 Illustrative Examples

A representative sample of AesthetiX-5K dataset is shown in Figure 1 which highlights the relationship of visual composition of light, texture, symbolism, and style with emotional interpretation. Each example is augmented with dominant emotion, and how perceived emotion is a contribution of color, lighting, texture, posture, and composition. Together, these examples provide qualitative insight into how emotions such as serenity, tension, and happiness are reflected through different artistic styles and visual structures. As shown in Figure 1, the following instances exemplify how visual style and composition collectively evoke different affective responses:

- Keisai Eisen's "Love on the Kamo River" is very serene. The calm and contemplative nature of the piece is reminiscent of the ukiyo-e art style.

- Pablo Picasso's "Bottle, Glass, and Violin" expresses tension. The use of fractured shapes and crossed planes reflects the discordant thought patterns of Synthetic Cubism.

- Monet's "Flower Arrangements" makes me feel happy. Soft brushstroke and color palette that is high key diffuse contours producing calm luminous character

## 3.4 Challenges During Annotation

Subjectivity is the biggest challenge in perception of emotion. Three primary difficulties were identified during annotation.

1. Divergent interpretation: the annotators sometimes perceived emotion of the art different from the artist's intent.

2. Cultural bias: Cultural background of annotators influenced their interpretation. For example, minimal composition were sometimes perceived melancholic vs serene.

3. Symbolic dependence: The presence of features/objects like halos, skulls and abstract figures required background knowledge for accurate interpretation.

These conflicts were mitigated by creating references that lined the formal elements of each image with corresponding emotional label.

---

[1]The annotators described emotions using a rich and diverse vocabulary, including terms such as *serenity*, *melancholy*, *tension*, and *awe*. These fine-grained emotions were mapped to six well known categories ensure the consistency taxonomy: *happiness*, *sadness*, *fear*, *anger*, *disgust*, and *surprise* to ensure consistency of model during training and evaluation. This normalization preserves the expressive richness of the human-written explanations while providing a structured and consistent label space for learning and evaluation.

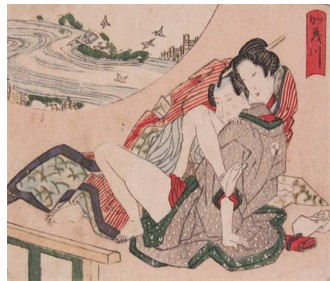 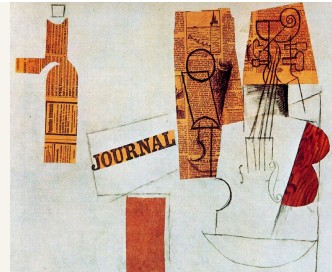 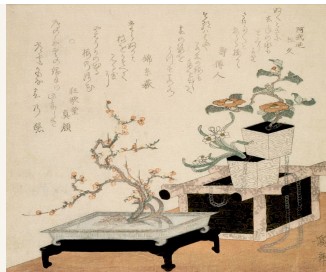

**Serenity**
*This painting evokes*:
Serenity

Muted indigo tones and symmetric reflections evoke calm companionship typical of ukiyo-e.

**Tension**
*This painting evokes*:
Tension

Fragmented geometry and intersecting planes suggest dissonance and cognitive strain.

**Happiness**
*This painting evokes*:
Happines

Soft brushwork and a high-key color palette with diffused contours produce luminous calm.

Figure 1: **Representative examples from AesthetiX-5K.** Each sample pairs the painting image with its annotated emotion and human-written explanation. From left to right: (a) Keisai Eisen – Serenity; (b) Pablo Picasso – Tension; (c) Claude Monet – Happiness.

### 3.5 Quality Control and Validation

Senior reviewers with expertise in art history re-evaluated a randomly selected 10% of the samples to verify the semantic correctness of the explanations.Additionally, a BERT-based textual coherence metric combined with manual proofreading was used to filter low-confidence annotations. To minimize digitization bias, each image was processed for color correction and normalized to a resolution of $512 \times 512$.

### 3.6 Ethical and Cultural Considerations

All images were sourced from open-license museum repositories such as Wikimedia Commons, The Metropolitan Museum of Art, and the Art Institute of Chicago. Annotations emphasize emotions as perceived rather than presumed ensuring that no moral or interpretive stance is attributed to the artist. AesthetiX-5K dataset consists diverse artistic traditions ranging from European modernism to Japanese woodblock prints and aims to promote inclusivity in cultural representation. The dataset aligns with the goals of SDG 10 (Reduced Inequalities) and SDG 16 (Peace, Justice and Strong Institutions) making digital cultural understanding accessible to all.

## 4 Proposed Model: AesthetiX-RAG

AESTHETIX-RAG is a neuro-symbolic model for *causally interpretable emotion recognition and grounded reasoning generation* in fine art. The architecture unifies a dedicated tri-branch visual encoder, an external Artist–Style–Motif–Emotion (ASME) knowledge graph, counterfactual style interventions for causal enforcement, and a retrieval-augmented generation (RAG) module for faithful explanation, see Figure 2.

### 4.1 Tri-Branch Visual Encoder for Disentangled Perception

Given an input image/painting $x$, three encoders, namely, Style Branch $(S(x))$, Content Branch $(C(x))$, and the Iconography Branch $(I(x))$ extracts respective representations $T_S$, $T_C$, and $T_I$, respectively. The final representation is created by concatenation of the three as $T = [T_S \| T_C \| T_I]$. It allows the model to capture different aspects of artistic perception that contribute to emotional interpretation. Following the description of each branch.

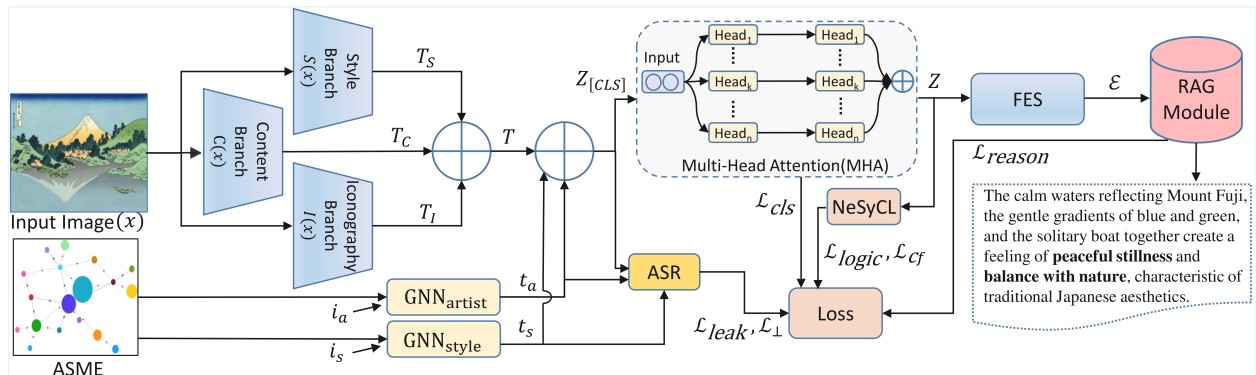

Figure 2: The AesthetiX-RAG Architecture

**Style Branch** ($S$): it extracts emotional features and low-level stylistic features and produces $T_S$ which contains information like brushstroke texture, color palette, and visual intensity. It is implemented via Vision Transformer Dosovitskiy et al. (2021) and wavelet-based color-histogram inspired by neural style analysis Gatys et al. (2016).

**Content Branch** ($C$): it captures the regional embedding $T_C$ that represents objects, scene layout and spatial relationship in the picture. It is implemented by Grounding-DINO Liu et al. (2024) which is a open-vocabulary detector.

**Iconography Branch** ($I$): this encoder models the conceptual and symbolic elements of the image related to artistic meaning. IconClass-style hierarchical representations Snell et al. (2017) is used to convert abstract information to token representations $T_I$ that support higher-level emotional reasoning.

### 4.2 Artist–Style–Motif–Emotion (ASME) Knowledge Graph

The ASME graph $\mathcal{G}$ formalizes established artistic relationships ({*Artist, Style, Motif, Emotion*} nodes) to supply structured contextual priors that often act as prediction shortcuts. **Contextual Embedding Generation.** For a painting linked to indices $i_a$ and $i_s$, two lightweight relational GNNs ( Schlichtkrull et al. (2018)) process the respective learnable node embeddings $\mathbf{h}_a^{(0)}$ and $\mathbf{h}_s^{(0)}$:

$$\mathbf{h}_a^{(1)} = \sigma(\mathbf{W}_{as}\mathbf{h}_s^{(0)} + \mathbf{W}_a\mathbf{h}_a^{(0)}), \tag{1}$$

$$\mathbf{h}_s^{(1)} = \sigma(\mathbf{W}_{sa}\mathbf{h}_a^{(0)} + \mathbf{W}_s\mathbf{h}_s^{(0)}), \tag{2}$$

yielding the fused prior $\mathbf{h}_{asme} = \mathrm{MLP}([\mathbf{h}_a^{(1)}; \mathbf{h}_s^{(1)}])$. This prior is projected into control tokens $t_a$ and $t_s$, concatenated to the visual sequence, and passed through the Multi-Head Attention (MHA):

$$Z = \mathrm{MHA}([T \,\|\, t_a \,\|\, t_s]),$$
$$\hat{y} = \mathrm{softmax}\left(W_o\left(Z_{[\mathrm{CLS}]} + \lambda_a t_a + \lambda_s t_s\right)\right). \tag{3}$$

**Anti-Shortcut Regularization.** To enforce learning from visual evidence $T$, we constrain the contextual tokens $(t_a, t_s)$:

1. Adversarial Leakage Penalty: Penalizes context tokens for easily predicting their protected attribute (artist/style) ( Ganin & Lempitsky (2015)):

$$\mathcal{L}_{\mathrm{leak}} = \rho_a \max(0, \mathrm{AUC}(h_a(t_a)) - \tau_a)$$
$$+\rho_s \max(0, \mathrm{AUC}(h_s(t_s)) - \tau_s). \tag{4}$$

2. Orthogonality Constraint: Encourages orthogonality between the stylistic subspace ($t_s$) and the semantic/prediction subspaces ($T_C$ and $Z_{[\text{CLS}]}$) to ensure concept disentanglement:

$$\mathcal{L}_\perp = \|\text{cov}(t_s, \text{pool}(T_C))\|_F + \|\text{corr}(t_s, W_o Z_{[\text{CLS}]})\|_F. \tag{5}$$

### 4.3 Neuro-Symbolic Consistency and Causal Interpretation

We define soft logical rules $r_\ell(Z)$ to embed art-theoretical knowledge (e.g., *Baroque ∧ chiaroscuro ⇒ solemnity*). The satisfaction degree is maximized via:

$$\mathcal{L}_{\text{logic}} = \sum_\ell \lambda_\ell \, \text{softplus}(-r_\ell(Z)). \tag{6}$$

Simultaneously, we enforce **causal interpretation** via targeted style-preserving perturbations $T_S'(x)$ (e.g., saturation modulation). The model is required to align its prediction shift with the expected emotional shift $\Pi_s$ for the intervention ( Kaushik et al. (2019)):

$$\mathcal{L}_{\text{cf}} = \left\| \sigma(f(T_S'(x))) - \Pi_s(\sigma(f(x))) \right\|_2. \tag{7}$$

Minimizing $\mathcal{L}_{\text{cf}}$ ensures that the style branch is a causal determinant of the final emotion.

### 4.4 Faithful Evidence Selection and Retrieval-Augmented Generation (RAG)

The explanation pipeline relies on Faithful Evidence Selection and Retrieval-Augmented Generation: **Evidence Selection ($\mathcal{E}$).** A sparse attention module identifies $K$ critical evidence tokens $\mathcal{E} \subset T$ whose removal maximizes prediction confidence drop:

$$\mathcal{E} = \arg\max_K \left( \Delta p(y|T) - \Delta p(y|T \setminus \mathcal{E}) \right).$$

Here, $y$ denotes the predicted emotion label (one of six affect categories), and $p(y|T)$ represents the model's confidence for emotion $y$ given the complete visual token sequence $T$. **RAG Pipeline.** A bi-encoder retriever fetches top-$k$ domain knowledge snippets $\mathcal{K}$ from curated **Artist Cards** and **Style Cards**. A cross-attention decoder then generates the explanation $E$ by conditioning on the CLS token, visual evidence $\mathcal{E}$, and retrieved knowledge $\mathcal{K}$:

$$E = \text{Decoder}(Z_{[\text{CLS}]}, \mathcal{E}, \mathcal{K}). \tag{8}$$

**Faithfulness Enforcement.** We impose citation-based losses (CC: Citation Coverage; APr: Attribution Precision) and a contrastive loss ($\mathcal{L}_{\text{contrast}}$) to penalize hallucination:

$$\mathcal{L}_{\text{expl}} = \text{NLL}(E, E^*) + \lambda_{\text{cc}}(1 - \text{CC}) + \lambda_{\text{apr}}(1 - \text{APr}) + \mathcal{L}_{\text{contrast}}. \tag{9}$$

**End-to-end Faithfulness Alignment** couples the reasoning decoder and classifier head by ensuring the necessity of evidence for the final prediction:

$$\mathcal{L}_{\text{faith}} = \text{KL}(p(y|T) \,\|\, p(y|T \setminus \mathcal{E})). \tag{10}$$

The overall join reasoning objective is $\mathcal{L}_{\text{reason}} = \mathcal{L}_{\text{expl}} + \mathcal{L}_{\text{faith}}$.

### 4.5 Overall Multi-Objective Optimization

The final multi-objective loss unifies all components for end-to-end optimization of perception, reasoning, and artistic cognition:

$$\begin{aligned} \mathcal{L} = \alpha \mathcal{L}_{\text{cls}} + \beta \mathcal{L}_{\text{reason}} + \eta \mathcal{L}_{\text{logic}} + \xi \mathcal{L}_{\text{cf}} \\ + \lambda \mathcal{L}_{\text{leak}} + \mu \mathcal{L}_\perp \end{aligned} \tag{11}$$

Here, each loss term enforces a distinct dimension of interpretability and hence ensuring model is validated across all axes under consideration.

We comprehensively evaluate AESTHETIX-RAG across quantitative, human, and qualitative dimensions. The proposed model achieves state-of-the-art accuracy and calibration while producing the most faithful, grounded, and interpretable emotional reasoning for fine art, establishing a new benchmark for affective visual analysis.

Table 1: Emotion classification performance on AesthetiX-5K.

| Model | Accuracy | Macro-F1 | AUC | ECE↓ |
|---|---|---|---|---|
| ResNet-50 ( He et al. (2016)) | 68.4 | 66.2 | 0.78 | 0.094 |
| ViT-B/16 ( Dosovitskiy et al. (2021)) | 70.6 | 68.1 | 0.80 | 0.086 |
| GalleryGPT ( Bin et al. (2024)) | 72.4 | 70.3 | 0.82 | 0.083 |
| OpenCLIP L/14 ( Cherti et al. (2023)) | 75.6 | 73.8 | 0.85 | 0.076 |
| ArtGraph ( Castellano et al. (2023)) | 74.9 | 72.5 | 0.84 | 0.076 |
| GIT-2 ( Wang et al. (2022)) | 75.2 | 73.1 | 0.86 | 0.073 |
| LLaVA-1.5 ( Liu et al. (2023)) | 78.4 | 76.9 | 0.85 | 0.074 |
| **AesthetiX-RAG (ours)** | **84.7** | **83.2** | **0.93** | **0.041** |

## 4.6 Emotion Classification Results

As shown in Table 1, accuracy and Macro-F1 steadily improve with deeper multimodal grounding, culminating in state-of-the-art performance achieved by the full AESTHETIX-RAG configuration. Across benchmarks (Table 1), our model outperforms all contemporary vision-language and vision-only baselines, achieving an $\approx$ 6-point gain in Accuracy and Macro-F1 over LLaVA-1.5, while attaining superior discrimination (AUC 0.93) and robustness.

Table 2: Comparative results on the ArtEmis and EmoArt-130k datasets

| Model | ArtEmis | | EmoArt-130k | |
|---|---|---|---|---|
| | Accuracy | Macro-F1 | Accuracy | Macro-F1 |
| BLIP-2 ( Li et al. (2023)) | 65.4 | 63.7 | 69.1 | 67.8 |
| GIT-2 ( Wang et al. (2022)) | 67.2 | 65.9 | 70.3 | 68.5 |
| LLaVA-2 ( Liu et al. (2023)) | 71.8 | 69.6 | 73.5 | 71.2 |
| **AesthetiX-RAG (ours)** | **76.9** | **75.1** | **79.6** | **77.9** |

Further, we conducted a comparative evaluation on two publicly available datasets, ArtEmis (Achlioptas et al. (2021)) and EmoArt-130k (Zhang et al. (2025)), to benchmark the generalization against representative multimodal baselines. Table 2 shows results on ArtEmis (Achlioptas et al. (2021)) and EmoArt-130k (Zhang et al. (2025)), where AESTHETIX-RAG consistently achieves the better performance among all compared methods. For ArtEmis, our model achieves 76.9% Accuracy and 75.1 Macro-F1, surpassing BLIP-2, GIT-2, and LLaVA-2 (71.8/69.6). For EmoArt-130k, AESTHETIX-RAG achieves 79.6% Accuracy and 77.9 Macro-F1, compared to LLaVA-2's 73.5% Accuracy and 71.2 Macro-F1. Overall, the proposed method achieves consistently higher Accuracy and Macro-F1 on both benchmarks, reflecting improved class-balanced emotion recognition under diverse,and improved robustness under diverse, large-scale data distributions.

## 4.7 Reasoning Quality and Faithfulness

We evaluate the generated explanations using both linguistic fluency and factual faithfulness metrics. Textual coherence is measured via BLEU ( Papineni et al. (2002)), ROUGE-L ( Lin (2004)), METEOR ( Denkowski & Lavie (2014)), and BERTScore ( Zhang et al. (2019)). Faithfulness metrics include Causal Consistency (**CC**) which is the alignment between visual and textual attention and the human-annotated emotional rationale, Attribution Precision (**APr**) it is defined as Phrase-level overlap with annotated human reasoning phrases, and Non-Hallucination Ratio (**NHR**) defined as proportion of generated explanations free from

Table 3: **Automatic reasoning quality and faithfulness metrics.** BLEU, ROUGE-L, METEOR, and BERTScore assess linguistic quality; Causal Consistency (CC), Attribution Precision (APr), and Non-Hallucination Rate (NHR) capture factual grounding and interpretive faithfulness. AESTHETIX-RAG consistently surpasses all baselines, producing explanations that are both fluent and causally grounded in retrieved art-historical context.

| Model | BLEU | ROUGE-L | METEOR | BERTScore | CC | APr | NHR |
|---|---|---|---|---|---|---|---|
| BLIP-2 ( Li et al. (2023)) | 0.32 | 0.41 | 0.33 | 0.84 | 0.52 | 0.48 | 0.71 |
| InstructBLIP (Vicuna-7B) ( Dai et al. (2023)) | 0.36 | 0.45 | 0.37 | 0.86 | 0.59 | 0.54 | 0.75 |
| GIT-2 ( Wang et al. (2022)) | 0.38 | 0.46 | 0.37 | 0.87 | 0.64 | 0.59 | 0.78 |
| LLaVA-1.5 (13B) ( Liu et al. (2023)) | 0.35 | 0.44 | 0.36 | 0.86 | 0.61 | 0.55 | 0.74 |
| **AesthetiX-RAG (ours)** | **0.48** | **0.61** | **0.49** | **0.92** | **0.87** | **0.81** | **0.94** |

unsupported or stylistically incorrect claims. As shown in Table 3, traditional captioning models (BLIP-2, GIT-2) achieve moderate lexical overlap scores but exhibit weaker causal grounding (CC ≈ 0.5–0.6).

As observed from Table 2 and Table 3 it can be observed that LLaVA-1.5 shows an improvement in contextual fluency but remains limited in factual precision. The proposed model **AesthetiX-RAG** show best overall performance on BERTScore, CC, APr, and NHR as 0.92, 0.87, 0.81, and 0.94, respectively. It demonstrates that **AesthetiX-RAG** is able to do causal reasoning and enhances both interpretability and linguistic quality.

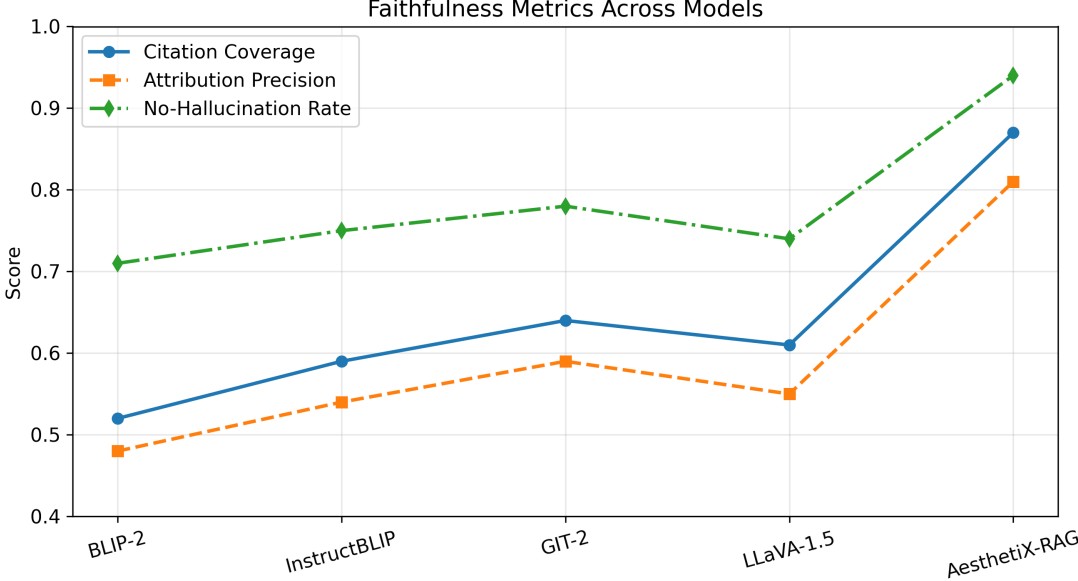

Figure 3: Faithfulness metrics (CC/APr/NHR) performance of proposed and competing models. AesthetiX-RAG is better in semantic explanation and least susceptible to hallucination.

**Faithfulness and Linguistic Trends.** As apparent from Figure 3, proposed model **AesthetiX-RAG** achieves better performance on CC, APr, and NGR showcasing that proposed framework is able to generate causally grounded and hallucination free interpretations aligned with visual evidence.

Additionally, Table 3 shows that there is a clear relationship inverse between perplexity and BLEU score, i.e., lower the perplexity better the quality of explanation demonstrated by high BLEU score. It suggests that causal reasoning and retrieval-guided contextual grounding jointly contribute to both accurate explanation and coherent language generation.

## 4.8 Human Evaluation

To evaluate the interpretability and perceptual quality of the generated explanations, we conducted a human evaluation study involving **4 domain experts** from fine arts (1), cognitive psychology (1), and artificial intelligence (2). The study included **2,400 annotated samples** ($4 \times 600$ randomly selected predictions) generated by BLIP-2, LLaVA, GIT-2, and AESTHETIX-RAG. Participants assessed each explanation using a 5-point Likert scale across three criteria: *Relevance*, *Faithfulness*, and *Interpretive Depth*. These are defines as follows.

- **Relevance**: It quantifies the explanation generated by a model alignment of emotional content with artwork.
- **Faithfulness**: It measures the quality of alignment between explanation and emotional content.
- **Interpretive Depth**: It measures the quality of depth of emotional reasoning.

A high inter-rater agreement was ($\kappa = 0.81$) observed which indicates indicates high level of consistency among annotators. The quantitative results presented in Table 4 show that AESTHETIX-RAG received the highest ratings across all evaluation dimensions, achieving scores of **4.4** for Relevance, **4.5** for Faithfulness, and **4.5** for Interpretive Depth. In comparison to AESTHETIX-RAG, the average baseline scores were 3.9, 3.7, and 3.5, respectively. These findings suggest that the proposed framework generates explanations that are not only linguistically coherent, but also closely aligned with the visual and stylistic characteristics of the artwork. In particular, the model demonstrates a stronger ability to relate artistic elements such as brushstroke patterns, tonal composition, and visual atmosphere to the emotions perceived by viewers.

Table 4: The results of Human evaluation on the Likert Scale from 1 to 5. The inter-rater agreement is $\kappa = 0.81$.

| Aspect (1–5) | BLIP-2 | LLaVA | GIT-2 | **AesthetiX-RAG** |
|---|---|---|---|---|
| Relevance | 3.7 | 3.9 | 4.0 | **4.4** |
| Faithfulness | 3.4 | 3.8 | 4.0 | **4.5** |
| Interpretive Depth | 3.1 | 3.6 | 3.8 | **4.5** |

## 4.9 Qualitative Analysis and Discussion

Table 5 presents qualitative comparisons of emotion understanding across representative paintings. Baseline models such as BLIP-2, LLaVA, and GIT-2 mainly generate descriptive captions focused on objects, scenes, or visible content, but often fail to explain how artistic form contributes to emotional meaning. Whereas AESTHETIX-RAG is able to produce detailed emotion-reason explanation. It suggests that AESTHETIX-RAG is able to connect visual and stylistic features conveyed by the art/picture. These explanations align more closely with human interpretation and established art-historical perspectives.

Across quantitative evaluations, human assessment studies, and qualitative analyses, AESTHETIX-RAG consistently demonstrates stronger performance than existing vision–language baselines in terms of accuracy, explanation faithfulness, calibration, and interpretive quality. Lower the perplexity higher is the quality and more contextually grounded is the explanation. As inferred from qualitative examples AESTHETIX-RAG is capable of relating stylistics characteristics and compositional structure for emotional interpretation and affective understanding. Overall, the results indicate that AESTHETIX-RAG provides a more interpretable and context-aware approach for emotion reasoning in fine art, supporting future research in culturally informed and human-centered multimodal AI systems.

## 4.10 Ablation

Finally, the ablation study (Table 6) shows that performance consistently decreases across all ablated variants, highlighting the importance of causal objectives, the ASME graph, and retrieval-augmented reasoning.

Table 5: Qualitative comparison of emotional inference and visual reasoning across baseline multimodal models and the proposed AESTHETIX-RAG. Each painting (left) is accompanied by emotion predictions and textual rationales from BLIP-2, LLaVA, and GIT-2 (middle), versus our causally grounded interpretation (right). Unlike the baselines—which produce surface-level affective tags derived from visual description—AESTHETIX-RAG synthesizes contextual features of color harmony, composition, and artistic intent to articulate the underlying emotional semantics of each artwork.

| Painting (Input) | Baselines (Emotion & Reasoning) | AesthetiX-RAG (Ours) |
|---|---|---|
| 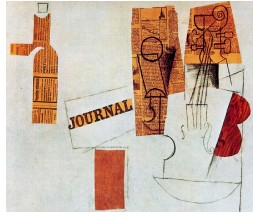 | **Emotion** 
 • *BLIP-2: Neutral* 
 • *LLaVA: Calm* 
 • *GIT-2: Peace* 
 **Reasoning:** 
 • *BLIP-2: Two people near a river.* 
 • *LLaVA: Couple standing by water, **calm mood**.* 
 • *GIT-2: A **peaceful** riverside scene.* | **Emotion: Serenity** 
 **Reasoning:** "Soft vermilion tones and balanced composition evoke calm companionship typical of Ukiyo-e domestic scenes." |
| 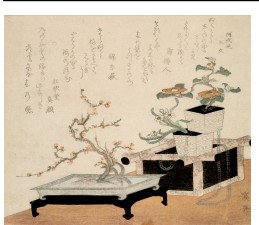 | **Emotion** 
 • *BLIP-2: Neutral* 
 • *LLaVA: Tension* 
 • *GIT-2: Neutral* 
 **Reasoning:** 
 • *BLIP-2: Abstract shapes of instruments.* 
 • *LLaVA: Cubist composition with **tension**.* 
 • *GIT-2: Still life in cubist style.* | **Emotion: Tension** 
 **Reasoning:** "Intersecting diagonals and fragmented planes disrupt spatial stability, producing cognitive strain characteristic of Synthetic Cubism." |
| | **Emotion** 
 • *BLIP-2: Neutral* 
 • *LLaVA: Neutral* 
 • *GIT-2: Neutral* 
 **Reasoning:** 
 • *BLIP-2: Woman arranging flowers.* 
 • *LLaVA: Person with vase indoors.* 
 • *GIT-2: Flower arranging scene.* | **Emotion: Happiness** 
 **Reasoning:** "Pastel palette, circular motion, and gentle posture evoke aesthetic joy associated with harmony in domestic Japanese art." |

Table 6: Ablation study of AESTHETIX-RAG. AesthetiX-RAG achieves near-perfect calibration, confirming confidence–accuracy alignment.

| Variant | Accuracy | Macro-F1 |
|---|---|---|
| **AesthetiX-RAG** | **84.7** | **83.2** |
| – RAG | 80.3 | 78.9 |
| – ASME Graph | 78.8 | 77.1 |
| – Anti-shortcut Reg | 76.2 | 74.4 |
| – Causal Loss $\lambda$ | 75.1 | 73.3 |
| – Retrieval Priors | 77.5 | 75.8 |

## 5 Conclusion

In this work, we introduced AESTHETIX-RAG, a retrieval-augmented multimodal reasoning framework designed to support emotion recognition and explanation in fine art. The framework combines a *tri-branch visual encoder* for disentangled representation learning, a *causal reasoning graph* that models relationships between artistic features and emotional interpretation, and a *knowledge-grounded RAG module* that generates explanations aligned with visual and contextual evidence. This enables, the proposed AESTHETIX-RAG to identify dominant emotion in the artwork along with apt explanation of emotional perception. To evaluate

the proposed model an annotated corpus textbfAesthetiX-5K dataset is curated. The results show that AESTHETIX-RAG demonstrates strong performance compared to baseline and competing models along with improved interpretability of artwork. Beyond empirical improvements, the framework highlights the importance of integrating contextual knowledge, stylistic reasoning, and explainable inference within affective computing systems. More broadly, this work moves toward a form of multimodal AI that does not simply describe visual content, but attempts to reason about how artistic structure and cultural context shape emotional understanding.

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
