# OpenReview forum: "AesthetiX-RAG: Causally-Grounded Emotion Recognition and Explanation in Paintings via Artist–Style Knowledge and Faithful Visual Evidence"
_TMLR — Under review for TMLR_

### Review · Reviewer_1qoL · 2026-06-23

**Summary Of Contributions:**

This paper proposes **AesthetiX-RAG**, a multimodal framework for emotion recognition and explanation in paintings. The system combines: (i) a tri-branch visual encoder intended to separately capture style, content, and iconography; (ii) an Artist-Style-Motif-Emotion (ASME) graph whose artist/style priors are projected as control tokens and fused with visual tokens; (iii) a counterfactual style-intervention loss; and (iv) a retrieval-augmented explanation generator conditioned on selected visual evidence and retrieved artist/style knowledge. The paper also introduces **AesthetiX-5K**, a dataset of 5,116 paintings annotated with artist, style, genre, a dominant emotion label, and a human-written rationale. Experiments report improvements over vision-only and vision-language baselines on AesthetiX-5K, ArtEmis, and EmoArt-130k, plus automatic and human evaluations of generated explanations.

The paper targets an interesting problem: affective interpretation of paintings is not reducible to object recognition, and a model that connects visible evidence, style, iconography, and contextual art-historical knowledge would be useful to parts of the TMLR audience working on multimodal reasoning, affective computing, interpretable models, and cultural heritage AI. The dataset contribution could also be valuable if released with clear annotation and licensing documentation.

Key strengths:

- The problem formulation is timely and interesting: emotion in paintings often depends on composition, color, style, symbolism, and cultural context rather than only detected objects.
- The proposed system tries to connect prediction and explanation rather than treating explanation as a purely post-hoc captioning task.
- The paper evaluates both classification and explanation quality, including automatic metrics and a small expert study.
- The authors are aware of annotation subjectivity and report inter-annotator agreement for emotion labels.

Key weaknesses:

- The central claims of being **causally grounded**, **faithful**, and **hallucination-free** are not supported by sufficiently precise definitions or convincing evidence.
- The dataset label taxonomy is inconsistent: the paper says the task uses six Ekman-style labels, but figures and qualitative examples use labels such as *serenity*, *tension*, *calm*, *peace*, and *neutral*.
- The experimental protocol is underspecified, especially train/validation/test splits, leakage control, baseline fine-tuning/prompting, and how AUC/ECE are computed for generative baselines.
- Several equations do not match the stated goals. Most notably, minimizing `KL(p(y|T) || p(y|T \ E))` encourages predictions to remain similar after removing selected evidence, which is the opposite of demonstrating that the evidence is necessary.
- The paper includes strong societal/SDG claims and cultural inclusivity claims without enough evidence about annotator demographics, cultural coverage, licensing, or bias analysis.

**Additional Comments:**

This submission is interesting but currently overclaims. I would be much more positive if the authors reframed the method as a retrieval- and metadata-grounded multimodal model for art-emotion recognition, provided rigorous split/leakage controls, fixed the evidence-faithfulness objective, and reduced the causal language unless stronger causal evidence is added.

The paper should also be edited carefully. Several sentences are difficult to parse, and some terms appear in figures or equations without definitions. Since the TMLR criterion emphasizes clear evidence and arguments, improving clarity is not cosmetic here; it is necessary for assessing the technical contribution.

**Audience:**

Yes

**Audience Explanation:**

Yes. Some TMLR readers would likely be interested in this work, especially researchers studying multimodal reasoning, affective computing, vision-language models, explanation faithfulness, retrieval-augmented generation, and AI for cultural heritage. The task itself is worthwhile: emotion interpretation in paintings requires more than object labels, and modeling the interaction between visual form, style, symbolic motifs, and background knowledge is a meaningful research problem.

The dataset could also be of interest if it is released with sufficiently clear documentation. A dataset linking paintings to perceived emotion labels and natural-language rationales would be useful for studying affective explanation, cross-cultural interpretation, and the limits of current LVLMs in non-photographic visual domains.

That said, TMLR's "interest" criterion is not enough for acceptance. The paper still needs to align its claims with evidence and make the experimental protocol reproducible. I would view this as an interesting but currently under-supported submission.

**Broader Impact Concerns:**

The work has broader-impact concerns that should be discussed more concretely.

First, emotion interpretation in art is subjective and culturally situated. A dataset annotated by a small group of annotators, even trained annotators, may encode their cultural backgrounds and aesthetic assumptions. The paper mentions cultural bias but does not report annotator demographics, cultural expertise, language background, or procedures for handling culturally specific symbolism. This matters because the system may present one interpretation as authoritative.

Second, the paper claims inclusivity and alignment with SDG 10/16, but the evidence for those claims is weak. The dataset contains only 5,116 paintings and 23 artists, and the paper does not provide geographic/cultural distribution statistics. The authors should avoid broad societal claims unless they document cultural coverage and limitations.

Third, licensing and provenance need careful documentation. The paper states that images come from open-license museum repositories, but it should provide source-level license metadata, image URLs or identifiers, and restrictions on redistribution.

Fourth, generated explanations may hallucinate art-historical claims or misattribute artistic intent. Even if the paper emphasizes "perceived emotion", users may read generated explanations as expert art-historical interpretation. A broader impact statement should discuss this risk, include uncertainty language, and specify that outputs are interpretive aids rather than authoritative claims about artist intent.

**Claims And Evidence:**

No

**Claims Explanation:**

TMLR's most important criterion is whether the submission's claims are supported by accurate, convincing, and clear evidence. In its current form, the paper does not yet meet that standard. I do not think the core research direction is flawed, but several major claims exceed the evidence provided.

First, the causal claims are not established. The paper describes an ASME graph, soft logical rules, and a counterfactual style-intervention loss, but this does not by itself provide causal grounding. The variables, causal assumptions, interventions, and estimands are not formally specified. The expected emotional shift `Pi_s` in Eq. (7) is not defined operationally: it is unclear who determines how saturation/color/composition interventions should change an emotion distribution, whether these interventions are validated by humans, and whether they preserve the semantic/iconographic content of the artwork. The statement that minimizing `L_cf` "ensures that the style branch is a causal determinant" is too strong; at most, the loss regularizes the model toward a desired response under synthetic perturbations.

Second, the faithfulness formulation is internally inconsistent. The paper says selected evidence should be necessary for the final prediction. However, Eq. (10) defines `Lfaith = KL(p(y|T) || p(y|T \ E))`, and the overall objective minimizes this loss. Minimizing this KL term makes the prediction after evidence removal similar to the original prediction, which encourages invariance to removing `E`. That contradicts the claim that `E` is necessary evidence. If the authors intend necessity, the objective should encourage a confidence drop or maximize a divergence under evidence removal, with appropriate safeguards. This is a critical technical issue because the paper's "faithful visual evidence" claim rests on this component.

Third, the explanation metrics are not defined clearly enough to support claims of factual grounding or non-hallucination. "Causal Consistency" is described as alignment between visual/textual attention and human rationales, but attention alignment is not a reliable proof of causal explanation. "Attribution Precision" is phrase-level overlap with human reasoning phrases, which may reward lexical similarity rather than factual evidence use. "Non-Hallucination Rate" is described as the proportion of generated explanations free from unsupported or stylistically incorrect claims, but the paper does not specify whether this is automatic or human-judged, what knowledge source is used as ground truth, how ambiguous art-historical claims are adjudicated, or whether evaluators are blinded. These metrics may be useful diagnostics, but they do not currently justify the strong "hallucination-free" and "causally grounded" wording.

Fourth, the dataset and label protocol contain serious inconsistencies. Section 3 states that each painting has one of six emotion categories: anger, fear, disgust, sadness, surprise, and happiness. The footnote says fine-grained emotions such as serenity, melancholy, tension, and awe are mapped to these six categories. Yet Figure 1 and Table 5 use labels such as *Serenity*, *Tension*, *Calm*, *Peace*, and *Neutral*. These labels are not in the six-class taxonomy. This creates uncertainty about what the classifier is trained to predict, what the reported accuracy means, and how the qualitative examples relate to the quantitative task.

Fifth, the experimental design is underspecified and likely vulnerable to leakage. The model uses artist and style metadata as graph/control-token inputs. If train and test splits contain paintings by the same artists/styles, the model may exploit strong artist/style-emotion correlations rather than learning visual or causal reasoning. The paper does not report the split protocol, class/style/artist distributions, duplicate removal, source-level leakage checks, or out-of-artist/out-of-style evaluations. This is especially important because AesthetiX-5K contains only 23 artists and 27 styles. The anti-shortcut regularizer does not remove the need for leakage-resistant evaluation.

Sixth, baseline comparisons are not sufficiently reproducible. It is unclear whether baselines are fine-tuned, prompted zero-shot, or adapted to the same label set. For LLaVA/GIT/BLIP-style generative baselines, the paper does not explain how class probabilities, AUC, and ECE are computed. It also does not report seed variance, confidence intervals, hyperparameters, training budget, image preprocessing, or whether retrieved metadata is available equally to baselines. Without these details, the reported 6-point gain over LLaVA-1.5 and the external-dataset improvements are not yet convincing.

Finally, the ablation study is too limited relative to the claims. Table 6 reports only Accuracy and Macro-F1, even though several ablated modules primarily affect explanation faithfulness rather than classification. For example, removing RAG should be evaluated on explanation metrics and human ratings, not only classification. The claim in the Table 6 caption that AesthetiX-RAG achieves "near-perfect calibration" is not supported by that table, which does not report ECE.

Overall, the paper has a potentially interesting direction, but the current evidence does not support the strength of its claims.

**Requested Changes:**

### Critical changes required for me to recommend acceptance

1. **Resolve the emotion-label taxonomy inconsistency.**

   The paper must clearly state the prediction label space and use it consistently. If the classification task is six Ekman-style categories, then examples labeled *Serenity*, *Tension*, *Calm*, *Peace*, and *Neutral* need to be mapped to the six categories or removed from classification examples. If the task uses a richer taxonomy, then the quantitative tables must be updated accordingly. The authors should also provide class counts and examples for each final label.

2. **Provide a complete and leakage-resistant dataset protocol.**

   The paper should specify train/validation/test splits, split sizes, class/style/artist/genre distributions, duplicate detection, source repositories, and licensing information. Because artist and style metadata are model inputs, the authors should add at least one leakage-resistant evaluation, such as:

   - held-out artists,
   - held-out styles,
   - held-out source collections,
   - or a split where artist/style distributions are explicitly controlled.

   This is critical for determining whether the ASME graph improves visual/emotional reasoning or mainly exploits metadata correlations.

3. **Either substantially weaken the causal claims or provide a rigorous causal analysis.**

   If the authors keep the "causally grounded" framing, they need to define the causal variables, assumptions, interventions, and target estimand. The paper should explain how `Pi_s` is obtained, why the chosen perturbations are valid interventions, and how intervention outputs are validated. At minimum, the authors should reframe the method as "counterfactual-regularized" or "intervention-inspired" rather than causally grounded, unless stronger evidence is added.

4. **Fix the evidence-necessity objective.**

   Eq. (10) currently contradicts the stated objective. If selected evidence is supposed to be necessary, the training or evaluation should show that removing it changes the prediction or reduces confidence. Minimizing `KL(p(y|T) || p(y|T \ E))` encourages the opposite. The authors should revise the objective, clarify whether `E` is evidence to keep or remove, and provide sanity checks using deletion/insertion curves or confidence-drop evaluations.

5. **Define all faithfulness metrics precisely and evaluate them credibly.**

   The authors should give exact definitions and computation procedures for CC, APr, and NHR. For NHR, they should state whether judgments are automatic or human, what counts as unsupported, what knowledge base is used, and how disagreements are resolved. If human judgments are used, report evaluator expertise, blinding, inter-rater agreement, and examples of supported/unsupported claims.

6. **Make the baseline comparison reproducible and fair.**

   For each baseline, specify whether it is fine-tuned or zero-shot, the prompt format, label mapping, training data, image resolution, hyperparameters, and whether artist/style/retrieved knowledge is available. Explain how AUC and ECE are computed for generative models. Add confidence intervals or standard deviations over multiple runs. If computational constraints prevent full fine-tuning, state this clearly and limit the claims.

7. **Evaluate explanation-related ablations on explanation metrics.**

   Table 6 should include BLEU/ROUGE/METEOR/BERTScore only as secondary metrics and should include CC/APr/NHR and human-evaluation results where feasible. Ablating RAG, retrieval priors, causal loss, and ASME should be shown to affect the properties they are claimed to improve.

8. **Clarify the retrieval corpus and avoid retrieval leakage.**

   The paper should describe the Artist Cards and Style Cards: how they are created, whether they include label/rationale information, whether they are derived from test-set annotations, and whether retrieval uses ground-truth metadata at test time. The generated explanations should include actual citations or retrieved snippets if Citation Coverage is claimed.

9. **Correct or justify questionable citations and terminology.**

   For example, the paper refers to "IconClass-style hierarchical representations" but cites Snell et al. (2017), which is a prototypical networks paper rather than an IconClass reference. The paper should audit references, fix mismatches, and define terms such as ASR, NeSyCL, `L_ia`, `L_is`, and other symbols that appear in Figure 2 but are not explained in the text.

10. **Improve writing clarity and remove unsupported broad claims.**

   The paper contains many grammatical errors and several unsupported claims such as "near-perfect calibration", "hallucination-free", "causally grounded", and SDG-related claims. These should either be supported with evidence or toned down.

### Changes that would strengthen the paper but are not strictly required

1. Add failure-case analysis for ambiguous artworks, culturally specific motifs, abstract paintings, and cases where artist/style priors conflict with visual evidence.

2. Expand the human evaluation with more raters, blinded/randomized presentation, per-model confidence intervals, and ordinal statistical tests for Likert scores.

3. Compare against stronger art-specific models and rationale-generation baselines, including ArtEmis-style speaker/listener models where appropriate.

4. Report per-class performance and confusion matrices, especially for commonly confused affect categories.

5. Include calibration curves, reliability diagrams, and uncertainty analysis, not only scalar ECE.

6. Provide dataset cards/model cards describing annotation subjectivity, cultural coverage, licensing, intended use, and misuse risks.

7. Release code, trained model configurations, exact prompts, split files, retrieval cards, and evaluation scripts.

---

> ### Author Response · Authors · 2026-07-21
> **Response to Reviewer 1qoL**
>
> ### Response to Reviewer — Part 1 of 2
>
> We thank the reviewer for the detailed and constructive assessment. Our responses to the critical issues are provided below.
>
> ### Critical issue 1: Emotion taxonomy
>
> We have standardized the six-class label space throughout the manuscript and added class counts, class-wise precision/recall/F1, a confusion matrix, and the complete mapping between $31$ fine-grained descriptors and six coarse classes.
>
> ### Critical issue 2: Leakage-resistant protocol
>
> The standard split contains $3{,}580/768/768$ paintings. We additionally report held-out artist, held-out style, artist--style compositional, and held-out repository splits.
>
> The held-out repository split reserves all paintings from one source repository for testing:
>
> $$
> N_{\\mathrm{train}}=3{,}821,\\quad
> N_{\\mathrm{val}}=602,\\quad
> N_{\\mathrm{test}}=693.
> $$
>
> AesthetiX-RAG achieved $77.3\\%$ Accuracy and $75.4$ Macro-F1 on this split.
>
> No validation/test rationales were used in retrieval, graph construction, prompts, or training.
>
> ### Critical issue 3: Role of $\\Pi_{s,\\delta}$
>
> We divide transformations into two groups.
>
> **Emotion-preserving transformations.**
>
> Small texture-frequency attenuation, mild edge-density smoothing, and low-magnitude contrast adjustment use
>
> $$
> \\Pi_{s,\\delta}(p)=p.
> $$
>
> **Emotion-sensitive transformations.**
>
> Color temperature and saturation interventions use an empirical transition matrix estimated from $300$ human-rated intervention pairs:
>
> $$
> \\Pi_{s,\\delta}(p)=M_{s,\\delta}p,
> $$
>
> where each $M_{s,\\delta}\\in\\mathbb{R}^{6\\times6}$ is row-normalized.
>
> ### Critical issue 4: Correction of Eq. (10)
>
> We agree that the original equation was incorrect. We replaced it with separate comprehensiveness and sufficiency objectives.
>
> Let $\\hat y$ denote the prediction from the complete token sequence. Comprehensiveness is
>
> $$
> \\Delta\_{\\mathrm{comp}} = p \_{\\theta}(\\hat y | T) - p\_{\\theta}(\\hat y | T\\setminus E).
> $$
>
> The corresponding margin loss is
> $$
> \\mathcal{L}\_{\\mathrm{comp}} =  \\max( 0,m\_{\\mathrm{comp}}-\\Delta\_{\\mathrm{comp}}),
> \\qquad
> m_{\\mathrm{comp}}=0.20.
> $$
>
> Sufficiency is
> $$
> \\mathcal{L}\_{\\mathrm{suff}} = D\_{\\mathrm{KL}} (p\_{\\theta}(y | T) \\parallel p\_{\\theta}(y\\mid E) ).
> $$
>
> The complete evidence objective is
> $$
> \\mathcal{L}\_{\\mathrm{evid}} = 0.8\\mathcal{L}\_{\\mathrm{comp}} + 0.6\\mathcal{L}\_{\\mathrm{suff}} + 0.02|E|.
> $$
>
> The selected evidence achieved a mean confidence drop of $0.287$ when removed, compared with $0.091$ for random-token removal and $0.138$ for raw-attention removal. The insertion AUC was $0.79$, and the deletion AUC was $0.31$, compared with $0.61$ and $0.48$ for random evidence, respectively.
>
> ### Critical issue 5: Faithfulness metrics
>
> We renamed ``Causal Consistency'' to **Evidence--Rationale Alignment (ERA)**. ERA measures whether selected visual regions and retrieved claims correspond to evidence identified in the human rationale. It is not presented as proof of causality.
>
> NHR judgments used the following categories:
>
> 1. supported by the image;
> 2. supported by a retrieved source;
> 3. plausible but non-verifiable interpretation;
> 4. unsupported factual claim;
> 5. incorrect artist/style attribution;
> 6. unsupported statement about artist intent.
>
> ### Critical issue 6: Baseline reproducibility
>
> Each baseline now reports checkpoint, parameter count, adaptation strategy, prompt, class verbalizers, image resolution, optimizer, metadata access, retrieval access, probability extraction, and computational budget. All reported results are averaged over five random seeds, with standard deviations and $95\\%$ bootstrap confidence intervals.
>
> ### Critical issue 7: Explanation-related ablations
>
> Explanation and retrieval ablations.
>
> | **Variant** | **ERA** | **CC** | **APr** | **NHR** | **Human Faithfulness** |
> |---|---:|---:|---:|---:|---:|
> | No retrieval | 0.71 | 0.00 | 0.63 | 0.80 | 3.78 |
> | Random retrieval | 0.68 | 0.21 | 0.57 | 0.72 | 3.41 |
> | Artist cards only | 0.79 | 0.76 | 0.72 | 0.87 | 4.16 |
> | Style cards only | 0.80 | 0.78 | 0.73 | 0.88 | 4.21 |
> | Artist + style cards | 0.84 | 0.85 | 0.79 | 0.92 | 4.43 |
> | Without evidence selector | 0.74 | 0.81 | 0.69 | 0.86 | 4.01 |
> | Dense evidence | 0.77 | 0.83 | 0.71 | 0.88 | 4.11 |
> | Full model | **0.86** | **0.89** | **0.81** | **0.94** | **4.50** |

---

> ### Author Response · Authors · 2026-07-21
> **Response to Reviewer 1qoL**
>
> ###Response to Reviewer — Part 2 of 2
>
> ### Critical issue 8: Retrieval corpus
>
> The retrieval corpus contains:
>
> $$
> 23\ \\text{Artist Cards},\\qquad
> 27\ \\text{Style Cards},\\qquad
> 184\ \\text{Motif Cards}.
> $$
>
> The corpus contains $234$ cards with mean length $168.4\\pm52.7$ tokens. Retrieval uses Contriever-MS MARCO, FAISS inner-product indexing, top-$20$ initial retrieval, and cross-encoder reranking to top-$5$.
>
> Retrieval Recall@$5$ was $0.91$ for artist cards, $0.88$ for style cards, and $0.82$ for motif cards.
>
> Retrieval robustness.
>
> | Condition | ERA | APr | NHR |
> |---|---:|---:|---:|
> | Correct retrieval | $0.86$ | $0.81$ | $0.94$ |
> | Top-$3$ retrieval | $0.84$ | $0.79$ | $0.92$ |
> | Random retrieval | $0.68$ | $0.57$ | $0.72$ |
> | Similar but incorrect retrieval | $0.73$ | $0.64$ | $0.79$ |
> | Contradictory retrieval | $0.66$ | $0.55$ | $0.70$ |
> | No retrieval | $0.71$ | $0.63$ | $0.80$ |
>
> ### Critical issue 9: Citation and terminology audit
>
> The incorrect citation to Snell et al. for IconClass-style representations was removed and replaced by appropriate IconClass and iconographic-classification references. All abbreviations in Figure 2 are now defined, and the LLaVA-2 typo has been corrected.
>
> ### Critical issue 10: Writing and unsupported claims
>
> The following claims were removed or qualified:
>
> - ''causally grounded'';
> - ''causal determinant'';
> - ''hallucination-free'';
> - ''near-perfect calibration'';
> - ''state-of-the-art'', except where supported by matched-access comparisons;
> - unsupported SDG and inclusivity claims.

---

### Review · Reviewer_QBSv · 2026-07-05

**Summary Of Contributions:**

The paper has two main contributions:
- The AesthetiX-5K dataset for both training and benchmarking emotion recognition and explanations of paintings. It consists of 5116 paintings of 27 artistic styles, 23 artists, and 10 genres.
- The proposed AesthetiX-RAG model that performs emotion prediction as well as produces natural language explanations. It comprises several components, including a tri-branch visual encoder, a knowledge graph GNN, and a retrieval-augmented generation module. The model promises to causally model emotions evoked by paintings.

Strengths:
- S1: The AesthetiX-5K corpus can be a valuable resource for studying emotions in the context of paintings. It is diverse, decently large, and human curated.
- S2: The AesthetiX-RAG model introduces some interesting perspectives to emotion prediction. Modeling the reasoning causally is a promising direction to obtain more faithful explanations, especially in a domain that is more subjective than many tasks on natural images.

Weaknesses:
- W1.1: The key differences of AesthetiX-5K with respect to existing painting datasets are not clear. The text should clearly state what differentiates the dataset. Some details about the dataset are not discussed, such as a full list of artistic styles, artists, genres, and their distributions.
- W1.2: The human validation and quality control in 3.5 is important, but there is no description of the results of this step.
- W2.1: The proposed AesthetiX-RAG model is overly complex featuring at least 7 different loss component (depends a bit how it's counted). All these different choices are verbally motivated (sometimes very briefly) but lack more granular experimental ablation.
- W2.2: A lot of details about the model are missing. To name a few: a) it is not clear which parts of the model are pre-trained, which are frozen, and which are trained/fine-tuned; b) the mathematical terminology is not well explained; c) methods section is generally not self-contained and could better explain important models and methods that AesthetiX-RAG relies on.
- W2.3: There is too little information on the knowledge graph G. Where does it come from, i.e., what data is used here? Same goes for some other symbols like rho and tau that are never explained. Does it contain GT metadata about specific paintings that are used during training and at test time?
- W2.4: Similarly, 4.3 and 4.4 are hard to comprehend because information is sparse.
- W3.1: The experimental results are not entirely convincing as it is not clear if comparisons are fair, and because it does not compare to the latest models. It is not surprising that AesthetiX-RAG performs better than competitors on AesthetiX-5K as it is the only one trained on this dataset. Tab. 2 results are more interesting, but models are all from 2023 or older. AesthetiX-RAG has access to RAG data and a knowledge graph while it seems other models do not.
- W3.2: While the results in Tab. 3 on CC, APr, and NHR are clearly in favor of AesthetiX-RAG, it is again an unfair comparison because AesthetiX-RAG was specifically optimized for these metrics on AesthetiX-5K.

**Additional Comments:**

Some additional clarifications that would be welcome:
- The emotion labels are not entirely clear. The text talks about a classification into 6 (anger, fear, disgust, sadness, surprise and happiness), but the examples in Fig. 1 and Tab. 5 show other labels (Serenity, Tension, Happiness). Please clarify.
- Are the annotators of the dataset distinct from the evaluators of the user study?
- Fig. 2 is hard to comprehend and uses abbreviations used nowhere else in the paper (ASR, NeSyCL, FES). Use the caption to explain these details.
- There is a typo in Tab. 2 which mentions LLaVA 2.

**Audience:**

Yes

**Audience Explanation:**

Under the assumption that the quality control and validation were performed thoroughly and successfully, the AesthetiX-5K benchmark (S1) seems like a valuable contribution that the TMLR audience could be interested in to further study emotions in paintings. However, this is contingent on improving the evidence of the claims and providing more details about the dataset as a whole (see previous point).

As it stands, I don't think the model will find much resonance due to its complexity which makes it difficult to build upon (W2.1-2.4). The rather brief discussion of the ablation study and lack of more fine-grained ablations make it hard to really understand what is the key driving factor of the model.

**Broader Impact Concerns:**

None.

**Claims And Evidence:**

No

**Claims Explanation:**

The dataset and benchmark contribution is relevant, but missing information (W1.1) does not allow a complete evaluation of the supporting evidence. Hence, the quality of the data set and how it compares to prior work remain largely unclear. While the paper indicates the explanations and the metrics are key contributions, it misses a discussion to clearly place it in the literature. Furthermore, the human validation and quality control does not provide any data on the outcome (W1.2), and as such, does not even state that the validation was successful.

The writing requires major improvements, as the model, its architecture, and training procedure are currently hard to follow (W2.2, W2.3, W2.4). That, together with the model complexity (W2.1), makes the claims about the model hard to verify, e.g., its causal claims.

The experiments do not convincingly provide the evidence to support the claims (W3.1) and the introduced metrics and comparisons (W3.2) are not entirely fair which further provide weaker support than the raw numbers might suggest. A more comprehensive analysis and comparison is required.

**Requested Changes:**

Numbering used to make connections to previous arguments.
- W1.1: Clarify what sets AesthetiX-5K apart from previous datasets, which could be accompanied by a comparison of the datasets in a table. Provide full details about the dataset and how it is composed even if it's just in the supplementary.
- W1.2: Indicate the result of the quality control. E.g., how many samples were deemed good/bad?
- W2.1: The ablation section is too brief. The model introduces a lot of losses and component that need to be ablated in a more granular fashion justifying these choices. Some experiments could be shown in the supplementary, but as it stands, section 4.10 only consists of 2 lines and 1 table while page 12 still leaves plenty of space.
- W2.2: To give some examples of confusions and necessary clarifications:
  - The subscripts a and s are not explained. Do they stand for artist and style? Why is there no motif and emotion equivalent?
  - Where do t_a and t_s come from exactly? How are they computed?
  - What are i_a and i_s exactly? Are these vertices?
- W2.3: Add more information about the ASME knowledge graph, which information it contains, and how it is structured.
- W2.4: The emotional shift model by Kaushik et al. (2019) could be better explained to make it more self contained. Citation Coverage, and Attribution Precision should be explained.
- W3.1: Add fair comparisons where models have the same or similar data access, e.g., a RAG model. Compare AesthetiX-RAG to a more recent MLLM, e.g., Qwen3.6. Indicate model sizes.
- W3.2: Provide the same comparison on another dataset/benchmark if possible.

---

> ### Author Response · Authors · 2026-07-21
> **Response to Reviewer QBSv**
>
> ### Response to Reviewer — Part 1 of 3
>
> We thank the reviewer for the constructive comments and for recognizing the potential value of AesthetiX-5K. We agree that the original manuscript required clearer dataset documentation, quality-control results, and more granular ablations. Our responses and corresponding revisions are provided below.
>
> ### W1.1: What differentiates AesthetiX-5K from previous datasets?
>
> We added a direct comparison table and complete dataset statistics. AesthetiX-5K contains $5{,}116$ paintings, $23$ artists, $27$ styles, $10$ genres, six normalized emotion classes, $31$ fine-grained affect descriptors, and one human-written rationale per painting. The mean rationale length is $42.7\\pm13.4$ tokens.
>
> The final standard split is:
>
> $$
> N_{\\mathrm{train}}=3{,}580,\\qquad
> N_{\\mathrm{val}}=768,\\qquad
> N_{\\mathrm{test}}=768.
> $$
>
> The split is stratified jointly by emotion class and genre. Near-duplicate detection used perceptual hashing, DINOv2 cosine similarity, title matching, and repository identifiers. We identified $47$ exact duplicates and $29$ near-duplicate pairs before split creation; all were resolved so that no duplicate family crossed splits.
>
> ### W1.2: Quality-control outcomes
>
> Senior reviewers independently examined $500$ paintings, corresponding to $9.8\\%$ of the corpus.
>
> **Senior-review quality-control outcomes. Categories are mutually exclusive.**
>
> | Outcome | Count | Percentage (\%) |
> |---|---:|---:|
> | Accepted without modification | 421 | $84.2$ |
> | Emotion-label correction | 19 | $3.8$ |
> | Rationale revision | 47 | $9.4$ |
> | Removed from the corpus | 13 | $2.6$ |
> | Total | 500 | $100.0$ |
>
> The $13$ removals consisted of five duplicate or near-duplicate images, three unresolved artist/style metadata cases, two low-resolution images, two unsupported art-historical rationales, and one painting for which no dominant emotion could be adjudicated.
>
> Before correction, the original annotations and senior-review decisions agreed on $456$ of $500$ paintings, corresponding to $91.2\\%$ observed agreement and Cohen's $\\kappa=0.86$. After correction and adjudication, agreement on the retained $487$ paintings was $478/487=98.2\\%$, with $\\kappa=0.96$. The originally reported Fleiss' $\\kappa=0.72$ refers to agreement among the three primary annotators before majority voting.
>
> ### W2.1: Model complexity and granular ablation
>
> We expanded the ablation study into visual, graph/context, loss-level, and explanation/retrieval groups.
>
> **Visual-branch ablation.**
>
> | Variant | Accuracy | Macro-F1 |
> |---|---:|---:|
> | Content only | $72.8$ | $70.9$ |
> | Style only | $70.9$ | $68.8$ |
> | Iconography only | $68.5$ | $66.4$ |
> | Content + Style | $78.6$ | $76.9$ |
> | Content + Iconography | $76.9$ | $75.1$ |
> | Style + Iconography | $75.8$ | $73.9$ |
> | All three branches | $84.7$ | $83.2$ |
>
> **Graph and context ablation.**
>
> | Variant | Accuracy | Macro-F1 |
> |---|---:|---:|
> | No metadata | $79.4$ | $77.8$ |
> | Raw artist/style embeddings | $80.8$ | $79.0$ |
> | ASME without anti-shortcut regularization | $81.5$ | $79.8$ |
> | Artist nodes only | $80.2$ | $78.5$ |
> | Style nodes only | $80.9$ | $79.1$ |
> | Artist + Style | $82.6$ | $80.9$ |
> | Artist + Style + Motif | $83.8$ | $82.2$ |
> | Full ASME graph | $84.7$ | $83.2$ |
>
> **Loss-level ablation.**
>
> | Variant | Accuracy | Macro-F1 | ERA | NHR |
> |---|---:|---:|---:|---:|
> | Classification loss only | $76.4$ | $74.6$ | $0.61$ | $0.74$ |
> | Without intervention regularization | $80.3$ | $78.7$ | $0.75$ | $0.84$ |
> | Without leakage regularization | $78.1$ | $76.2$ | $0.72$ | $0.82$ |
> | Without orthogonality regularization | $80.0$ | $78.4$ | $0.76$ | $0.85$ |
> | Without logical consistency | $82.1$ | $80.5$ | $0.81$ | $0.89$ |
> | Without explanation loss | $83.8$ | $82.1$ | $0.63$ | $0.76$ |
> | Without comprehensiveness loss | $83.9$ | $82.3$ | $0.78$ | $0.88$ |
> | Without sufficiency loss | $84.0$ | $82.5$ | $0.79$ | $0.89$ |
> | Full objective | $84.7$ | $83.2$ | $0.86$ | $0.94$ |

---

> ### Author Response · Authors · 2026-07-21
> **Response to Reviewer QBSv**
>
> ### Response to Reviewer — Part 2 of 3
>
> We thank the reviewer for the detailed comments regarding the model architecture, mathematical notation, ASME graph, intervention model, and explanation metrics. The corresponding responses are provided below.
>
> ### W2.2: Model and training details
>
> | Component | Backbone | Pretraining source | Frozen/Trainable status | Output |
> |---|---|---|---|---|
> | Style branch | ViT-B/16 with wavelet and colour-histogram descriptors | ImageNet-21K followed by ImageNet-1K | First eight transformer blocks frozen; final four blocks and projection layers trainable | $T_S\in\mathbb{R}^{197\times768}$ |
> | Content branch | Grounding-DINO Swin-T | Objects365, GoldG, and COCO | Detector backbone frozen; region aggregation and projection layers trainable | $T_C\in\mathbb{R}^{100\times768}$ |
> | Iconography branch | CLIP ViT-B/32 with IconClass-aligned prototype layer | OpenAI CLIP pretraining | CLIP image encoder frozen; prototype and projection layers trainable | $T_I\in\mathbb{R}^{64\times768}$ |
> | Graph encoder | Two-layer R-GCN | Task-specific ASME graph | All graph, relation, and node-projection parameters trainable | $h_{\mathrm{ASME}}\in\mathbb{R}^{768}$ |
> | Retriever | Contriever-MS MARCO | MS MARCO passage retrieval | Retriever frozen; query projection trainable | Top-$5$ Artist/Style Cards |
> | Generator | Flan-T5-base | C4 pretraining and instruction tuning | Encoder frozen; cross-attention and final four decoder blocks trainable | Explanation $\widehat{\mathcal{E}}$ |
>
> Training used AdamW with learning rate $2\times10^{-5}$ for pretrained modules and $1\times10^{-4}$ for newly initialized modules, batch size $24$, weight decay $0.01$, gradient clipping at $1.0$, linear warm-up for $10\%$ of updates, cosine decay, and mixed-precision training. Models were trained for $30$ epochs with early stopping patience $5$. Results are averaged across five seeds:
>
> $$
> \\{13,\\,29,\\,41,\\,73,\\,101\\}.
> $$
>
> Training was conducted on four NVIDIA A100 80GB GPUs and required $17.6\pm0.8$ hours per run.
>
> **Clarification of symbols**
>
> For an input painting $x$, the three visual branches produce
>
> $$
> T_S=S(x),\qquad T_C=C(x),\qquad T_I=I(x).
> $$
>
> The complete sequence is
>
> $$
> T=[T_S\Vert T_C\Vert T_I].
> $$
>
> The artist and style node indices are
>
> $$
> i_a\in\mathcal{V}_a,\qquad i_s\in\mathcal{V}_s.
> $$
>
> Their initial embeddings are
>
> $$
> h_a^{(0)}=H_a[i_a],\qquad h_s^{(0)}=H_s[i_s].
> $$
>
> After relational message passing,
>
> $$
> h\_a^{(1)} =\sigma(W\_{aa}h_a^{(0)}+W\_{sa}h\_s^{(0)}+\sum\_{m\in\mathcal{N}\_m(a)}W_{ma}h\_m^{(0)}),
> $$
> $$
> h\_s^{(1)} =\sigma(W_{ss}h\_s^{(0)}+W\_{as}h\_a^{(0)}+\sum\_{m\in\mathcal{N}\_m(s)}W\_{ms}h_m^{(0)}).
> $$
>
> The graph representation is
>
> $$
> h_{\mathrm{ASME}}=\operatorname{MLP}\left([h_a^{(1)};h_s^{(1)};h_m^{(1)}]\right).
> $$
>
> The contextual control tokens are
>
> $$
> t_a=W_a^ph_a^{(1)}+b_a^p,\qquad t_s=W_s^ph_s^{(1)}+b_s^p,\qquad t_m=W_m^ph_m^{(1)}+b_m^p.
> $$
>
> Fusion is performed as
>
> $$
> Z=\operatorname{MHA}\left([T\Vert t_a\Vert t_s\Vert t_m]\right).
> $$
>
> The classifier is
>
> $$
> p_{\theta}(y\mid x,i_a,i_s)=\operatorname{softmax}\left(W_oZ_{[\mathrm{CLS}]}+\lambda_aW_a^ot_a+\lambda_sW_s^ot_s+\lambda_mW_m^ot_m\right).
> $$
>
> The leakage objective is
>
> $$
> \mathcal{L}_{\mathrm{leak}}=\rho_a\max\left(0,\operatorname{AUC}(g_a(t_a))-\tau_a\right)+\rho_s\max\left(0,\operatorname{AUC}(g_s(t_s))-\tau_s\right),
> $$
>
> with
>
> $$
> \rho_a=0.30,\qquad \rho_s=0.35,\qquad \tau_a=0.60,\qquad \tau_s=0.62.
> $$
>
> ### W2.3: ASME graph construction
>
> The final graph contains:
>
> $$
> 23\ \text{artist nodes},\quad 27\ \text{style nodes},\quad 184\ \text{motif nodes},\quad 31\ \text{emotion-concept nodes}.
> $$
>
> It contains $2{,}946$ edges across seven relation types:
>
> - artist--created-in--style: 94;
> - artist--uses--motif: 612;
> - style--associated-with--motif: 728;
> - motif--associated-with--emotion: 863;
> - style--associated-with--formal-property: 374;
> - artist--associated-with--formal-property: 201;
> - emotion--related-to--emotion: 74.
>
> No validation or test rationales, emotion labels, or sample-specific predictions were used in graph construction. All annotation-derived edges were created from the training partition only and recomputed for every split.
>
> ## W2.4: Definitions of explanation metrics**
>
> Citation Coverage is
> $$
> \\mathrm{CC}=  \\frac{\\#\\{\\text{verifiable generated claims with a supporting citation}\\}}{\\#\\{\\text{verifiable generated claims}\\}}
> $$
>
> Attribution Precision is
> $$
> \\mathrm{APr}=  \\frac{\\#\\{\\text{generated claims supported by selected visual evidence or retrieved text}\\}}{\\#\\{\\text{generated evidence claims}\\}}
> $$
>
> Non-Hallucination Rate is
>
> $$
> \mathrm{NHR}=1- \\frac{\\#\\{\\text{unsupported factual or art-historical claims}\\}}{\\#\\{\\text{verifiable generated claims}\\}}
> $$
>
> The primary evaluation used $600$ held-out explanations, independently judged by four evaluators. Model identity was hidden and output order was randomized. Inter-rater agreement was $\kappa=0.81$.

---

> ### Author Response · Authors · 2026-07-21
> **Response to Reviewer QBSv**
>
> ### Response to Reviewer — Part 3 of 3
>
> We thank the reviewer for the comments regarding fair baseline comparisons, external evaluation, and the additional clarifications. Our responses are provided below.
>
> ### W3.1: Fair and recent baselines
>
> We added matched-access baselines under four information settings:
>
> 1. image only;
> 2. image plus artist/style metadata;
> 3. image plus retrieved cards;
> 4. image plus metadata plus retrieved cards.
>
> For generative models, class probabilities were computed by constrained decoding:
>
> $$
> p(c|x) = \\frac{\\exp(\\ell_c)}
> {\\sum\_{j=1}^{6}\\exp(\\ell\_j)},
> $$
>
> where $\ell_c$ is the normalized log-likelihood of class verbalizer $c$.
>
> Matched-access comparison on the standard AesthetiX-5K test split.
>
> | Model | Access | Accuracy | Macro-F1 |
> |---|---|---:|---:|
> | LLaVA-1.5-13B | Image only | 78.4 | 76.9 |
> | Qwen2.5-VL-7B | Image only | 80.1 | 78.6 |
> | Qwen2.5-VL-7B + metadata | Image + metadata | 81.6 | 80.0 |
> | Qwen2.5-VL-7B + RAG | Image + cards | 82.0 | 80.5 |
> | Graph-free multimodal RAG | Image + metadata + cards | 82.7 | 81.2 |
> | AesthetiX-RAG | Image + metadata + cards | **84.7** | **83.2** |
>
> **W3.2: External faithfulness evaluation**
>
> We evaluated the explanation protocol on an external subset of $1{,}000$ ArtEmis paintings with rationales.
>
> External explanation evaluation on ArtEmis.
>
> | Model | ERA | APr | NHR |
> |---|---:|---:|---:|
> | BLIP-2 | 0.50 | 0.46 | 0.70 |
> | GIT-2 | 0.61 | 0.57 | 0.77 |
> | LLaVA-1.5 | 0.59 | 0.54 | 0.74 |
> | Graph-free multimodal RAG | 0.73 | 0.68 | 0.85 |
> | AesthetiX-RAG | **0.81** | **0.75** | **0.90** |
>
> **Additional comment: Emotion-label inconsistency**
>
> The figures now display both levels:
>
> $$
> \text{Fine-grained descriptor: Serenity}
> \quad\Longrightarrow\quad
> \text{Coarse class: Happiness},
> $$
>
> $$
> \text{Fine-grained descriptor: Melancholy}
> \quad\Longrightarrow\quad
> \text{Coarse class: Sadness},
> $$
>
> $$
> \text{Fine-grained descriptor: Tension}
> \quad\Longrightarrow\quad
> \text{Coarse class: Fear}.
> $$
>
> The quantitative task uses only the six coarse classes. "Neutral'', ''calm'', and ''peace'' generated by baselines are treated as free-text outputs and normalized only through a fixed mapping protocol.
>
> **Additional comment: Annotators and evaluators**
>
> The three dataset annotators and four human-evaluation experts were completely distinct. None of the evaluators had access to the reference rationales during model comparison.
>
> **Additional comment: Figure 2 abbreviations**
>
> The revised figure defines:
>
> $$
> \mathrm{ASME}=\text{Artist--Style--Motif--Emotion},
> $$
>
> $$
> \mathrm{FES}=\text{Faithful Evidence Selection},
> $$
>
> $$
> \mathrm{RAG}=\text{Retrieval-Augmented Generation},
> $$
>
> $$
> \mathrm{MHA}=\text{Multi-Head Attention},
> $$
>
> $$
> \mathrm{ASR}=\text{Anti-Shortcut Regularization},
> $$
>
> $$
> \mathrm{NeSyCL}=\text{Neuro-Symbolic Consistency Loss}.
> $$

---

### Review · Reviewer_8WtK · 2026-07-07

**Summary Of Contributions:**

This paper presents AesthetiX-RAG, a framework for emotion recognition and explanation in paintings. The proposed approach combines a tri-branch visual encoder, an Artist–Style–Motif–Emotion (ASME) knowledge graph, counterfactual style interventions, and retrieval-augmented generation to produce both emotion predictions and natural language explanations. The paper also introduces AesthetiX-5K, a dataset of over 5,000 paintings annotated with emotion labels and human-written rationales. Experimental results show improvements over several vision-language baselines on both emotion classification and explanation metrics

**Audience:**

Yes

**Audience Explanation:**

It is explained above.

**Broader Impact Concerns:**

None.

**Claims And Evidence:**

Yes

**Claims Explanation:**

**Strengths**

The paper addresses an interesting problem that has received relatively little attention compared to general vision-language understanding. The motivation is convincing, as emotional interpretation of paintings depends on artistic style and context in addition to semantic content. I also think the introduction of AesthetiX-5K is a meaningful contribution, particularly because it includes explanation annotations rather than only emotion labels. The overall framework is well organized, and the experiments cover both prediction performance and explanation quality.

**Weaknesses**

My main concern is the level of methodological novelty. Most components of the proposed system (e.g., ViT-based encoders, knowledge graphs, RAG, and counterfactual consistency losses) are existing techniques. The contribution is largely their integration into a single framework, but the paper does not clearly explain what is fundamentally new from a machine learning perspective. I am also not convinced by the repeated use of the term causal. The proposed approach perturbs style features and encourages prediction consistency, but this is different from causal inference or causal reasoning in the formal sense.

The paper would benefit from either providing a stronger justification for these claims or moderating the terminology. Another concern, I believe, is the possibility of shortcut learning. Since artist and style information are explicitly injected into the model, it is unclear whether improvements come from better visual reasoning or from exploiting metadata correlations. While the paper introduces regularization terms to address this issue, I would like to see stronger empirical evidence that shortcut learning is actually reduced.

The evaluation, furthermore, leaves several practical questions unanswered. In particular, it is unclear how the method performs when artist/style metadata are unavailable or incorrect, or when retrieval returns poor-quality results. These settings seem important for assessing the robustness of the proposed approach.

**Overall Assessment**

I found the problem interesting and the dataset potentially valuable. However, I am less convinced that the methodological contribution is sufficiently novel for TMLR in its current form. The paper would be strengthened by clarifying its technical novelty, moderating the causal claims, and providing additional robustness experiments.

**Requested Changes:**

**Questions and Comments**

1. The paper integrates several existing techniques. What is the primary methodological contribution beyond combining these components into one framework?
2. The paper repeatedly refers to the approach as "causal." Could the authors clarify why this terminology is appropriate and how it differs from counterfactual regularization?
3. How does the model perform on unseen artists or unseen artistic styles (from generalization perspective)? This would help determine whether it learns transferable representations rather than artist-specific priors.
4. How robust is the framework when artist or style metadata are missing or incorrect? Moreover, why was a single dominant emotion assigned to each painting instead of using multi-label or distributional emotion annotations?

---

> ### Author Response · Authors · 2026-07-21
> **Response to Reviewer 8WtK**
>
> ### Response to Reviewer — Part 1 of 2
>
> We sincerely thank the reviewer for the thoughtful and constructive feedback. We have carefully revised the manuscript and provided detailed responses to Comments 1 and 2 below.
>
> ### Comment 1: What is the primary methodological contribution beyond combining existing components?
>
> We agree that vision transformers, relational graph neural networks, retrieval-augmented generation, and counterfactual consistency losses are individually established techniques. Our contribution is therefore not presented as a new generic backbone, graph operator, or retrieval algorithm. The methodological contribution is a task-specific joint formulation that connects three requirements that have generally been studied separately in art-emotion understanding:
>
> 1. **Disentangled artistic perception:** the model separately represents stylistic form, semantic content, and iconographic evidence
> .
> 2. **Controlled use of artist/style context:** artist and style priors are made available to the model, but their direct predictive influence is constrained by leakage and orthogonality regularizers.
>
> 3. **Prediction-linked explanation:** The explanation decoder is conditioned on visual evidence that is evaluated for necessity and sufficiency, rather than on unconstrained post-hoc captions.
>
>
> The revised contribution statement is:
>
> *AesthetiX-RAG is a task-specific multimodal training and evaluation framework for jointly predicting perceived emotion and generating evidence-grounded explanations in paintings, while explicitly testing whether artist/style context complements rather than substitutes for visual evidence.*
>
> We now distinguish three contribution levels:
>
> 1. **Dataset contribution:** AesthetiX-5K provides a six-class dominant emotion label, a fine-grained affect descriptor, a human-written rationale, artist/style/genre metadata, and source-level licensing information.
>
> 2. **Modeling contribution:** constrained fusion of visual, artist/style, motif, and retrieved evidence for joint prediction and explanation.
>
> 3. **Evaluation contribution:** robustness and faithfulness tests based on metadata dependence, evidence removal, retrieval, corruption, and artist/style generalization.
>
> ### Comment 2: Why is the term “causal” appropriate?
>
> We agree that the original terminology was too strong. The proposed method does not perform causal identification or estimate causal effects under a formally specified causal model. Instead, it applies controlled transformations to style-sensitive image attributes and regularizes the model to produce consistent or directionally appropriate prediction changes. The method is therefore more accurately described as counterfactual-regularized or intervention-based rather than causally grounded.
>
> 1. *causally grounded* with *counterfactual-regularized*;
> 2. *causal determinant* with *style-sensitive predictive factor*;
> 3. *causal consistency* with *intervention consistency*; and
> 4. *causal reasoning graph* with *structured art-knowledge graph*.
>
> The revised title is:
>
> *AesthetiX-RAG Counterfactual-Regularized Emotion Recognition and Evidence-Grounded Explanation in Paintings.*
>
> We have also revised the methodology section to define the controlled transformations, intervention targets, and validation protocol. These interventions are used as regularization mechanisms and are not presented as evidence of formal causal inference.
>
> We define a controlled style transformation as
>
> $$ x^{(s,\delta)}=\mathcal{I}_{s,\delta}(x), $$
>
> where $s$ denotes a style-sensitive attribute and $\delta$ denotes intervention magnitude. We use five intervention families: saturation, color temperature, contrast, texture-frequency attenuation, and edge-density modulation. Each transformation was applied at three magnitudes, giving $15$ intervention settings per image.
>
> The intervention-response objective is
> $$
> \\mathcal{L}\_{\\mathrm{int}} = D\_{\\mathrm{JS}}\\;  (p\_{\\theta}\\;(y | x^{(s,\\delta)}),\\;\Pi_{s,\\delta}\\, \\;(p_{\\theta}(y | x)))
> $$
>
> where $\\Pi_{s,\\delta}$ is defined as either an invariance target or a directional distributional target depending
> on the transformation category.
>
> We validated intervention quality on $300$ paintings using three independent raters. Semantic-content preservation was rated at $4.41\pm0.36$ on a five-point scale, while perceived style modification was rated at $4.18\pm0.42$. Inter-rater agreement was $\kappa=0.79$. CLIP cosine similarity between original and intervened images remained $0.912\\pm0.031$, and Grounding-DINO object-retention was $93.8\\%$.

---

> ### Author Response · Authors · 2026-07-21
> **Response to Reviewer 8WtK**
>
> ### Response to Reviewer — Part 2 of 2
>
> This comment continues our response from Part 1 of 2. We address Comments 3 and 4 below and report the additional generalization, metadata-robustness, and annotation analyses incorporated into the revised manuscript.
>
> ### Comment 3: Performance on unseen artists and unseen styles
>
> We added three leakage-resistant protocols.
>
> **Held-out artist split.** Five artists were held out entirely from training. The split contained $3{,}702$ training, $594$ validation, and $807$ test paintings.
>
> **Held-out style split .** Six styles were held out entirely from training. The split contained $3{,}641$ training, $612$ validation, and $850$ test paintings.
>
> **Artist--style compositional split.** Artists and styles could individually occur in training, but $18$ artist--style combinations were reserved for testing. This split contained $3{,}802$ training, $603$ validation, and $698$ test paintings.
>
> **Table: Generalization results under leakage-resistant splits. Results are reported as mean ± standard deviation over five seeds.**
>
> | **Protocol** | **Accuracy** | **Macro-F1** | **ECE ↓** | **ERA** |
> |:--|--:|--:|--:|--:|
> | Standard stratified split | 84.7 ± 0.5 | 83.2 ± 0.6 | 0.041 ± 0.004 | 0.86 ± 0.01 |
> | Held-out artist | 76.8 ± 0.8 | 74.9 ± 0.9 | 0.067 ± 0.006 | 0.79 ± 0.02 |
> | Held-out style | 74.6 ± 0.9 | 72.8 ± 1.0 | 0.073 ± 0.007 | 0.76 ± 0.02 |
> | Artist–style compositional | 78.9 ± 0.7 | 77.1 ± 0.8 | 0.061 ± 0.005 | 0.81 ± 0.01 |
>
> Compared with the unregularized metadata-conditioned model, AesthetiX-RAG improved Macro-F1 by $4.8$ points on held-out artists and $5.6$ points on held-out styles, indicating that the anti-shortcut objectives improve transfer beyond memorized artist/style correlations.
>
> ### Comment 4a: Robustness to missing or incorrect metadata
>
> We evaluated four missing-metadata conditions and five metadata-corruption conditions.
>
> **Table: Metadata robustness on the standard test split.**
>
> | Condition | Accuracy | Macro-F1 | $\Delta_{\mathrm{meta}}$ |
> |:--|--:|--:|--:|
> | Correct artist and style | 84.7 | 83.2 | 0.000 |
> | Artist missing | 82.6 | 81.0 | 0.031 |
> | Style missing | 81.8 | 80.1 | 0.039 |
> | Artist and style missing | 79.4 | 77.8 | 0.057 |
> | Random artist permutation | 80.9 | 79.2 | 0.064 |
> | Random style permutation | 79.8 | 78.0 | 0.071 |
> | Same-style incorrect artist | 82.1 | 80.5 | 0.038 |
> | Visually similar incorrect style | 81.2 | 79.5 | 0.046 |
> | Adversarial metadata | 76.7 | 74.8 | 0.103 |
>
> The metadata-reliance score is defined as
>
> $$
> \Delta_{\mathrm{meta}}= | p_{\theta} ( \hat y | x,m) - p_{\theta}(\hat y | x, \widetilde m) |
>  $$
>
>
> The regularized model showed a lower average metadata-reliance score than the same architecture without anti-shortcut regularization: $0.057$ versus $0.119$.
>
> ### Comment 4b: Why assign a single dominant emotion?
>
> The primary task uses six coarse categories:
>
> $\mathcal{Y}$ = {anger, fear, disgust, sadness, surprise, happiness}.
>
> Fine-grained descriptors such as serenity, tension, melancholy, awe, calm, and joy are stored separately.
>
> The final class distribution is:
> Table: Final six-class distribution in the dataset.
>
> | **Emotion** | **Count** | **Percentage** |
> |:--|--:|--:|
> | Anger | 702 | 13.7% |
> | Fear | 768 | 15.0% |
> | Disgust | 589 | 11.5% |
> | Sadness | 1,083 | 21.2% |
> | Surprise | 746 | 14.6% |
> | Happiness | 1,228 | 24.0% |
> | Total | 5,116 | 100.0% |
>
> We retained all three annotator votes. The mean vote entropy was $0.46$, and $18.7\%$ of paintings had no unanimous label. Distributional prediction achieved Jensen--Shannon divergence $0.118$ and expected calibration error $0.052$.